# Controlled condensation by liquid contact-induced adaptations of molecular conformations in self-assembled monolayers

Guoying Bai ®[1] ✉, Haiyan Zhang[1], Dong Gao[2], Houguo Fei[3], Cunlan Guo[3], Mingxia Ren[1] & Yufeng Liu[1]

Surface condensation control strategies are crucial but commonly require relatively tedious, time-consuming, and expensive techniques for surface-chemical and topographical engineering. Here we report a strategy to alter surface condensation behavior without resorting to any molecule-type or topographical transmutations. After ultrafast contact of liquids with and removal from surfaces, the condensation rate and density of water droplets on the surfaces decrease, the extent of which is positively correlated with the polarity of the liquid and the duration of contact. The liquid contact-induced condensation rate/density decrease (LCICD) can be attributed to the decrease of nucleation site density resulted from the liquid contact-induced adaption of surface molecular conformation. Based on this, we find that LCICD is applicable to various surfaces, on condition that there are flexible segments capable of shielding at least part of nucleation sites through changing the conformation under liquid contact induction. Leveraging the LCICD effect, we achieve erasable information storage on diverse substrates. Furthermore, our strategy holds promise for controlling condensation of other substances since LCICD is not specific to the water condensation process.

Gaining insights into the surface properties influencing water condensation and exploring strategies for controlling condensation on surfaces are crucial to wide-ranging fields, from fundamental research to industrial applications[1-5]. During the past century, surface condensation control strategies have predominantly focused on surface chemical and topographical engineering[2,6-17]. For example, self-assembled monolayer (SAM) patterns with heterogeneous surface free energies have been extensively constructed to control the spatial distribution of condensed water droplets[6,8,9,18,19]. However, chemical/topographical heterogenization engineering is always complicated and difficult to scale up because they have to use relatively tedious, time-consuming, and expensive techniques such as ion etching, laser

ablation, stamping, selective plasma etch, or photolithographic lift-off to achieve biphilic/topographical pattern[5,20].

Here, by a simple liquid-contact method, we realize the patterning of condensed droplets on surfaces modified by a single molecule type. The mechanism can be ascribed to the liquid contact-induced change of surface molecular conformation, which leads to the change of nucleation site density. Our findings expand the repertoire of surface condensation control strategies beyond palpable chemistry and topography heterogenization to encompass more subtle molecule conformation heterogenization. This enhances our understanding of water condensation mechanisms and facilitates improved spatial control of water condensation in numerous applications[2,6,7,15,21,22].

[1]Tianjin Key Laboratory of Materials Laminating Fabrication and Interface Control Technology, School of Materials Science and Engineering, Hebei University of Technology, Tianjin 300401, P. R. China. [2]Key Laboratory of Hebei Province for Molecular Biophysics, Institute of Biophysics, School of Health Science & Biomedical Engineering, Hebei University of Technology, Tianjin 300401, P. R. China. [3]College of Chemistry and Molecular Sciences, Wuhan University, Wuhan, Hubei 430072, P. R. China. ✉e-mail: baiguoying@iccas.ac.cn

## Results and discussion

### Water CICD effect

Taking a fluorinated SAM (1*H*,1*H*,2*H*,2*H*-perfluorodecyltrichlorosilane on a silicon wafer, Si-FDTS, see Methods for the preparation and Suppl. Fig. 1 for the characterization results) as an example, we observed the condensation process of water on the surface areas that were in contact with (referred to as "in") and without (referred to as "out") the liquid of water first (Methods). The condensation processes on other relatively hydrophilic SAMs (3-glycidoxypropyl(dimethoxy)methylsilane and 3-aminopropyltriethoxysilane on silicon substrate, Si-GOPTS and Si-APTES) exhibit similar behavior, and the results are shown in Suppl. Fig. 2.

As shown in Fig. 1a, a drop of water was first dripped onto the Si-FDTS surface and then immediately removed by rinsing the Si-FDTS with ethanol (EtOH) and blow-drying it with $N_2$ to ensure the surface was free from contaminations. Subsequently, the treated Si-FDTS was enclosed in a closed cell containing water droplets to provide humidity. Afterward, the cell was first heated to a temperature 5 °C

above room temperature to make the water droplets evaporate and then cooled to make the vapor condensate. Observing through an optical microscope equipped with a digital camera, we found that the density of condensed water droplets ($\rho$) in the area "in" was significantly lower than that in the area "out". This phenomenon is obvious under various investigated supersaturation values (Suppl. Figs. 3 and 4). Further examination of the entire condensation process (Fig. 1b and Suppl. Movie 1) reveals that the condensations of water droplets on the two areas start to occur simultaneously; but as time progresses, the $\rho$ in area "out" increases at a much faster rate compared to that in area "in" and both eventually reach the maximum values at the same time of about 177 s (Fig. 1c). In contrast, the size of a droplet in area "out" increases significantly slower compared to that in area "in", resulting in the maximum diameter of a droplet in area "in" is nearly twice that in area "out" (Fig. 1d). The contact angles of water on the two areas show no significant difference (Suppl. Fig. 5), suggesting that the observed size difference of condensed water droplets in the two areas is due to the real difference of droplet volumes. This

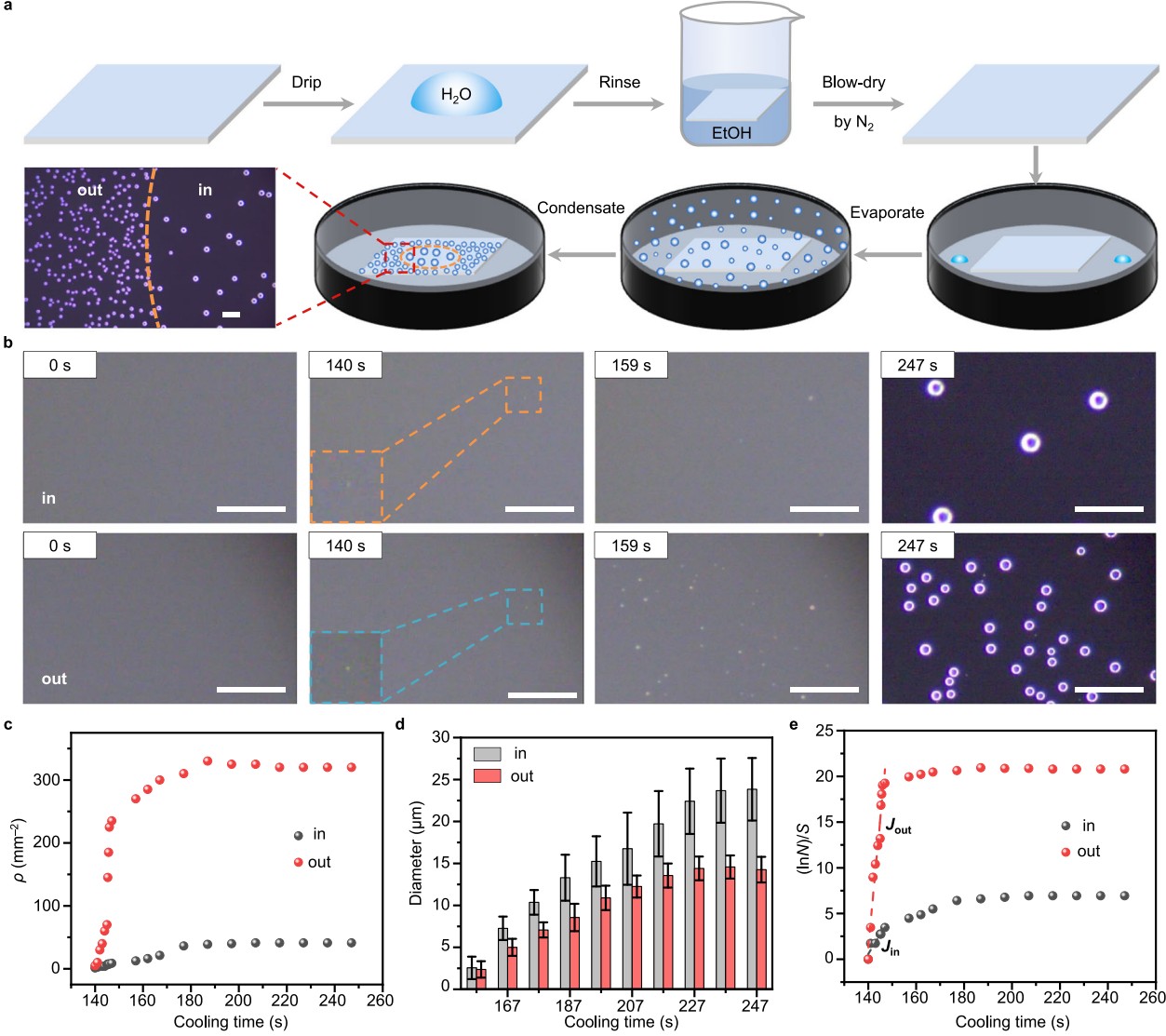

**Fig. 1 | Water contact-induced condensation behavior change. a** Schematic diagram of the sample pretreatment and condensation experiment process. **b** Optical microscopic images (dark-field) showing typical condensation processes of water in the area "in" and "out" of the Si-FDTS surface. Time 0 represents the moment when the sample temperature starts to cool down. The insets zoom in on the initially condensed water droplets. **c** Variation of $\rho$ in area "in" and "out" with

cooling time. **d** Variation of condensed droplet diameter in area "in" and "out" with cooling time. The error bars are standard deviations based on three measurements. **e** Relationship between the number of condensed droplets $N$ in area "in"/"out" and the cooling time. The slope of the initial stage reflects the initial steady-state nucleation rate. All the scale bars are 100 μm.

difference is reasonable because the lower $\rho$ makes the local supersaturation around each droplet in the area "in' relatively higher and thus contributes to the faster growth of the droplet[23–25]. The total water droplet volumes in the area "in" and "out" are also quantified (Suppl. Fig. 6).

To quantificationally compare the nucleation rates of water droplets in the two areas, we plotted the relationship between the number of condensed droplets ($N$) and the cooling time ($t$) based on the equation[26] of

$$\ln(N) = J S t \qquad (1)$$

where $J$ is the nucleation rate of water droplets, and $S$ is the surface area of the observed zone. As shown in Fig. 1e, by fitting the slope of the initial stage of the curve (ln $N$)/$S$ versus $t$, we can obtain the initial steady-state nucleation rate. It is pronounced that $J_{out}$ is larger than $J_{in}$, consistent with the faster increase of $\rho$ in area "out".

## Mechanism of water CICD

To elucidate the mechanism behind the water CICD effect, we first conducted a series of experiments to exclude the possible reasons, such as the change of surface chemistry (Suppl. Figs. 5 and 7)/topography (Suppl. Fig. 8) and adsorption of water molecules (Suppl. Figs. 5 and 9)/volatile organic compounds (Suppl. Figs. 10 and 11) induced by water contact. In addition, the condensation behavior difference between the areas "in" and "out" maintains when the two areas are separated and placed in separate chambers for condensation (Suppl. Fig. 12), excluding the possibility of mutual condensation interferences between the two areas. Furthermore, we find that the water CICD effect doesn't apply to the unmodified silicon surface (Suppl. Fig. 13), but remains on SAMs, irrespective of whether the substrate is flat or nanostructured, hard or soft (Suppl. Figs. 14 and 15).

Based on these experimental results, it is evident that the SAM molecules on the surface play a key role in the water CICD effect. Given that the conformation of SAM molecules can be influenced by liquid[27,28], we investigated whether the conformation of SAM molecules changed upon water contact by using polarization modulation infrared reflection absorption spectroscopy (PMIRRAS), which is a valuable analytical technique for characterizing organic (sub-)monolayer structures on reflecting surfaces (mostly metal surfaces)[27,29,30]. In addition, to acquire desired target peaks that do not overlap with the irrelevant signals, such as the signals of substrates or non-target chemical bonds, an alkanethiol (1-dodecanethiol) SAM on Au surface (Au−C$_{12}$, see Methods for preparation) was used as a research model. It exhibits the same water CICD effect as that observed on various silane SAMs (Fig. 2a, b). As shown in Fig. 2d, e, the PMIRRAS spectra of Au-C$_{12}$ reveal characteristic stretching modes of methyl and methylene groups, namely $v_{as}(CH_3)$, $v_{as}(CH_2)$, $v_s(CH_3)$, and $v_s(CH_2)$ at 2964, 2921, 2878, and 2851 cm$^{-1}$, respectively[31,32], where the intensity of $v_s(CH_3)$ or $v_{as}(CH_3)$, i.e., a function of the projection of the corresponding transition dipole moment along the surface normal, can reflect the tilt of the hydrocarbon chain (Suppl. Fig. 16)[33]. With chain tilt increasing, the intensity of $v_s(CH_3)$ increases, while $v_{as}(CH_3)$, being orthogonal to $v_s(CH_3)$, exhibits a decreasing intensity[34]. To assess the conformational change of Au-C$_{12}$ after water contact induction, we calculated the intensity ratio between $v_s(CH_3)$ and $v_{as}(CH_3)$. Interestingly, we observed an approximate 7% increase of the $Iv_s(CH_3)/Iv_{as}(CH_3)$ ratio after water contact treatment, suggesting a statistical increase trend of the hydrocarbon chain tilt. More PMIRRAS data are shown in Suppl. Fig. 17.

Looking back on the condensation process, we know that condensation occurs simultaneously in the areas "in" and "out" (Fig. 1b and Suppl. Movie 1). This suggests that the nucleation-free energy barrier of water condensation remains unchanged after water contact induction because of the fact that condensation preferentially occurs on

surfaces with lower nucleation-free energy barriers under the same kinetically driving force (e.g., the environment conditions for condensation such as humidity and temperature, etc.)—for example, the preferential condensation on the hydrophilic surface (possessing lower water nucleation free energy barrier compared with the hydrophobic surface) rather than on the hydrophobic one (Suppl. Movie 2) and the spatially controlled water droplet array formed on the hydrophilic spots[8]. Based on the nucleation rate equation

$$J = \rho_S Z A_{kin} \exp(-\Delta G^* / (k_B T)) \qquad (2)$$

with parameters being the number of nucleation sites per unit area/volume, $\rho_S$, the Zeldovich factor (accounting for the probability that a postcritical cluster will grow rather than dissolve), $Z$, the kinetic prefactor (the atomic or molecular mobility of the prenucleated phase), $A_{kin}$, the free energy barrier for nucleation, $\Delta G^*$, Boltzmann constant, $k_B$, and absolute temperature, $T$, the water contact-induced condensation rate difference and the resulting apparent condensation density difference can be ascribed to the change of $\rho_S$ due to the constant $Z$, $A_{kin}$, and $T$ for the condensation of the same substance on the congeneric samples under the identical condensation conditions in our experiment and the unaltered $\Delta G^*$ as mentioned above.

According to the experimental findings that water contact induces the increase of SAM hydrocarbon chain tilt and the decrease of nucleation site density, the mechanism of the water CICD effect can be proposed as Fig. 2g, h. It is important to note that experimentally prepared SAMs are not perfect and typically contain defects, which can serve as favorable nucleation sites during the water condensation process due to the relatively higher hydrophilicity of the exposed substrate surface at these defective sites compared to the SAMs themselves. This can be confirmed by the increasing condensation density on surfaces with decreasing FDTS molecular density, i.e., Si-FDTS with increasing defects (Suppl. Fig. 18). When SAMs come into contact with water, the molecular chains become more tilted, which can be seen as a response of the SAMs' minimizing their available free volume to prevent permeation of water into the bulk of the relatively hydrophobic monolayers[27]. Consequently, the more tilted chains can cover some of the original surface defects, resulting in a decreased nucleation site density and, thus, a decreased condensation rate/density.

This mechanism can be further consolidated by the simultaneous recovery trend of the SAM molecular chain tilt and the condensation rate/density. As shown in Fig. 2c,f, after the Au-C$_{12}$ that has been previously contacted with water is soaked in EtOH for a long period (e.g., 24 h here), the $\rho$ on it returns to a similar level as that on the initial surface without water contact, and so does the ratio of $Iv_s(CH_3)/Iv_{as}(CH_3)$, suggesting the recovery of the molecular chain tilt. The correlation between the molecular conformation and condensation behavior can be visualized as Fig. 2i. After soaking in EtOH, the water contact-induced tilted chains restore towards the initial state, which can be explained by the SAM chains being in a more fluid environment and capable of restructuring themselves' conformation under the infiltration and solvation of EtOH (contact angle of EtOH on Au−C$_{12}$ < 5°)[27]. The recovery of the chain tilt makes the defects, i.e., the favorable nucleation sites, exposed again and thus leads to the increased condensation rate/density.

## Liquid and surface applicability of LCICD effect

Based on the above findings, we speculate that the other liquids beyond water may also make the condensation rate/density decrease through contact induction, and this LCICD effect may be applicable on more diverse surfaces beyond SAMs, provided that liquid contact induces changes in surface molecular conformation and consequently reduces nucleation site density. Therefore, we systematically explored the contact induction effects of various liquids with different polarities

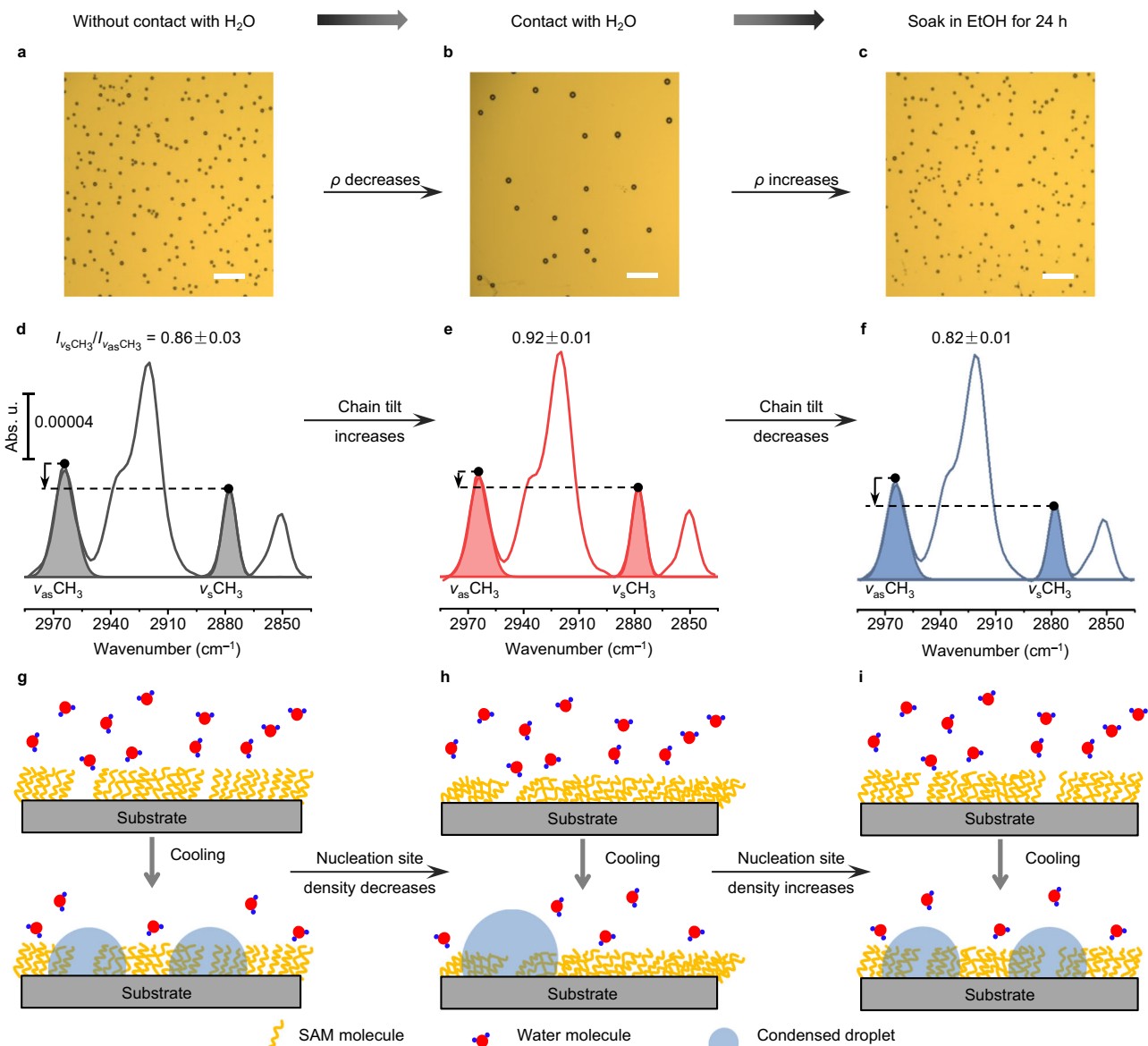

**Fig. 2 | Mechanism for the water CICD effect. a–c** Typical optical microscopic images (bright-field) of the condensed water droplets on Au–$C_{12}$ that have been sequentially pretreated without water contact, with water contact, and with EtOH soaking. Note that these images show the condensation state in the equilibrium stage (i.e., both the density and diameter of the condensed water droplet reach the maximum). All the following condensation images are captured in the equilibrium stage of condensation unless particularly stated. All the scale bars are 100 µm. **d–f** Typical PMIRRAS spectra of Au–$C_{12}$ that have been sequentially pretreated without water contact (gray), with water contact (red), and with EtOH soaking (blue). The ratios of $I_{v_s}(CH_3)/I_{v_{as}}(CH_3)$ (mean ± standard error on the mean) are given in the figure. **g–i** Schematic diagram of the mechanisms for the water CICD and the EtOH soaking-induced condensation rate/density recovery effect.

on water condensation behaviors (Fig. 3a, b) and the applicability of LCICD on surfaces with structure segments of varying degrees of flexibility (Fig. 3c).

As shown in Fig. 3a, the ratios of $\rho_{out}/\rho_{in}$ (the $\rho$ here represents the droplet density in the equilibrium stage of condensation) and $J_{out}/J_{in}$ are used to quantify the degree of LCICD; ratios >1 indicate the presence of LCICD, with higher ratios indicating stronger LCICD degrees, where Si-FDTS is taken as a surface example for condensation. It is evident that the variations of $\rho_{out}/\rho_{in}$ and $J_{out}/J_{in}$ with liquid type/liquid-surface contact duration show the same trend. For strong polar liquid–water, a prominent LCICD is observed, with the degree of LCICD maximized at a transient contact duration achievable experimentally (e.g., 5 s). For other relatively weak polar liquids, the degree of LCICD is much weaker than that induced by water and shows a slight increase trend with contact duration increasing and reaches maximum when the contact duration is extended to about 360 s; in addition, despite

minor differences, the degree of LCICD induced by various liquids follows a decreasing tendency in the order of DMSO, DMF, EtOH, and PE, particularly when the contact duration exceeds 120 s, which order is consistent with the polarity decreasing order. These results indicate the dependence of LCICD degree on liquid polarity and contact duration (Fig. 3b). Both increasing liquid polarity and extending contact duration within limits contribute to the LCICD effect. This can be rationalized by considering that the liquid of stronger polarity should elicit a more pronounced response of SAMs (Fig. 2) to minimize their contact with the liquid due to the increasing repellency of SAMs towards liquids of increasing polarity[27] (Suppl. Fig. 19) and thus lead to the greater decrease of nucleation site density. Similarly, extending contact duration allows for a longer period of stimulation of liquids to SAM surfaces, thus reinforcing the LCICD effect.

Figure 3c shows the condensation differences between the area "in" and "out" on a series of surfaces with structure segments of varying

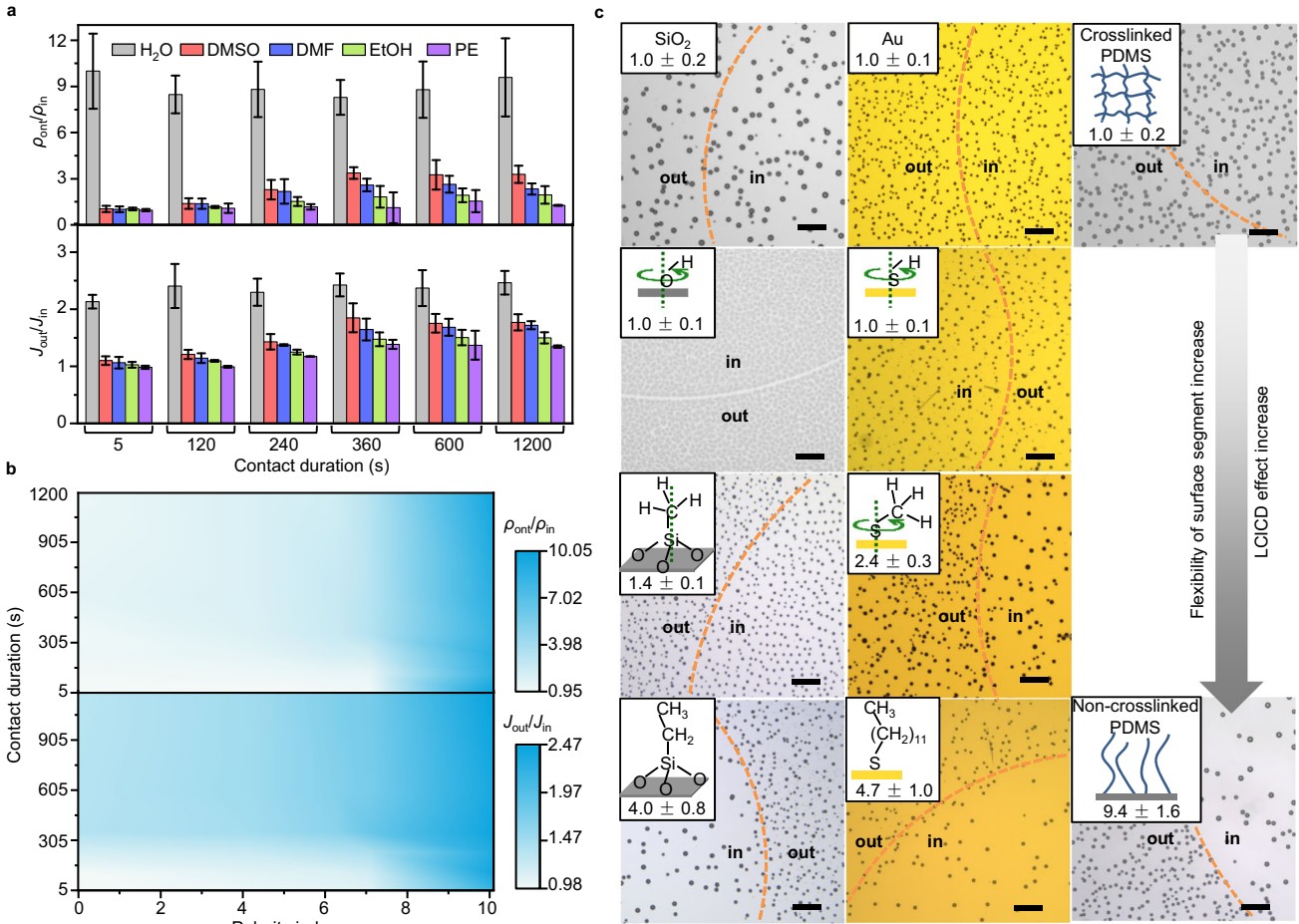

**Fig. 3 | Liquid and surface applicability of LCICD effect. a** Degree of LCICD (quantified by $\rho_{out}/\rho_{in}$ and $J_{out}/J_{in}$) under the induction of water, dimethyl sulfoxide (DMSO), N,N-dimethyl formamide (DMF), EtOH, and petroleum ether (PE) with varying contact durations. The error bars are standard deviations on three measurements. **b** Heat maps representing the degree of LCICD plotted as a function of the liquid polarity and the contact duration. The polarity index here is a relative value (the polarity index of butyl chloride is 1). **c** Typical optical bright-field microscopic images of the condensed water droplets in area "in" and "out" of a series of surfaces with structure segments of various degrees of flexibility. The insets show the segment structures and the values of $\rho_{out}/\rho_{in}$. All the scale bars are 100 μm.

degrees of flexibility. Surfaces lacking rotatable bonds, such as SiO₂ and Au, exhibit identical condensation density in the areas "in" and "out" ($\rho_{out}/\rho_{in} = 1.0$), indicating the absence of LCICD effects. When these surfaces are modified with simple groups like -OH and -SH, which possess a certain degree of orientation freedom with O/S rotating but are insufficient to cover surface defects through their orientation changes, inhomogeneous condensation occurs, albeit still without LCICD effects. On hydroxylated SiO₂ or O₂ plasma-activated Au surface (Suppl. Fig. 20), a narrow gap of 5–10 μm devoid of condensation at the "in"–"out" area boundary is observed. This can be attributed to the reorientation of initially disorder-oriented -OH groups at the "in"–"out" area boundary towards the area "in" after water contact induction, facilitated by hydrogen bonding between water and -OH. Consequently, the "in"–"out" area boundary becomes nucleating-inert compared to other areas due to the loss of randomly oriented -OH groups' shielding the SiO₂/Au substrate (note that the hydroxylated SiO₂ or oxygen radical-rich Au surface exhibits higher condensation nucleation activity than the SiO₂/Au substrate). A similar reorientation of -SH groups may also occur on the thiolated Au surface following water contact induction. However, no gap is observed in this case, which is likely due to the relatively sparse distribution of condensed droplets. When surfaces are modified with relatively larger segments such as methyl silane (Si–C₁) or methyl mercaptan (Au–C₁), LCICD effects appear, with Au–C₁ demonstrating a more pronounced effect due to

the non-linear Au–S–C bond-leaded the greater freedom of -CH₃. Theoretically, Si–C₁, where the Si–C bond is perpendicular to the surface ideally, and the whole segment cannot tilt, should not exhibit an LCICD effect. The very weak LCICD effect observed experimentally can be explained by the commonly imperfect grafting of silane on substrate—incomplete Si–O condensation (Suppl. Fig. 21)[35], making the Si–C bond capable of swinging. As the modification further extends to longer flexible chains, e.g., alkyl chains containing more than two carbon atoms, the LCICD effect becomes more prominent, as shown in the images of droplets condensed on Si substrates modified with ethyl silane (Si–C₂) or Au–C₁₂.

As for the flexible polymers such as polydimethylsiloxane (PDMS) of varying degrees of chain freedom (see Methods for preparation details), similar findings are obtained (right column of Fig. 3c). Specifically, no LCICD effect is observed on the crosslinked PDMS networks; while a noticeable LCICD effect is observed on the non-crosslinked PDMS brushes. Additionally, we note that on the crosslinked PDMS networks having not been washed to remove the unreacted PDMS oligomers, LCICD also occurs (Suppl. Fig. 22), further consolidating the importance of chain freedom to the liquid contact-induced adaption of surface molecular conformation and thus the LCICD effect. Based on the above experimental results, the prerequisite for surfaces exhibiting LCICD effect can be concluded—the surfaces should contain flexible segments that are sufficient to cover/reduce at least a portion

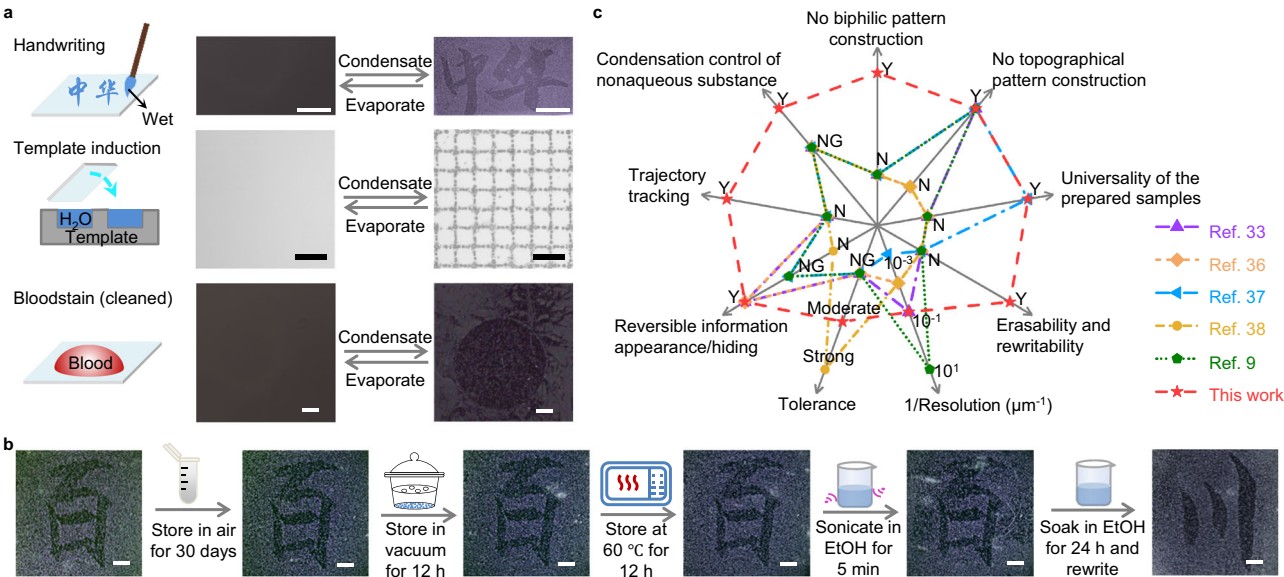

**Fig. 4 | Application of the LCICD effect for information storage. a** Sketch maps of information storage strategies and typical optical microscopic images showing rapid and reversible information appearance/hiding under water condensation/evaporation. **b** Typical optical microscopic images showing the stability of the stored information after the sample was treated under various conditions and the erasability of the stored information after the sample was soaked in EtOH for 24 h. The scale bar in a is 100 μm. All the other scale bars are 1 mm. **c** Comparison of this work with previously reported condensation control strategies in terms of sample preparation and applications. "Y," "N," and "NG" denote "yes", "no", and "not given" in the literature, respectively. The universality of the prepared samples means that our method achieves the separation of the information storage process from the sample preparation process. Therefore, we need not customize the special samples according to the information to be stored. That is, the prepared samples are applicable for any information to be stored.

of the originally exposed favorable nucleation sites, i.e., the relatively low-nucleation-energy-barrier sites such as the more hydrophilic defects, through changing their conformations under the liquid contact induction.

## Application of the LCICD effect for information storage

Based on the LCICD effect, it becomes possible to distinguish areas that have been in contact with liquids from those that have not by observing the distribution of condensed water droplets. This discovery inspires us to develop a way for information storage. As shown in Fig. 4a, Chinese character or pattern information can be easily stored on SAM surfaces by simple handwriting or template induction. Moreover, we can successfully track the trajectory of liquid-containing systems such as blood (which has been cleaned) on surfaces even though the contact duration is instantaneous. All the information can be readily observed by simply breathing upon the surfaces. Furthermore, the stored information can be erased and rewritten through a special method that can refresh the surface molecular conformation, such as soaking in EtOH for a long time, although it shows excellent atmospheric, vacuum, thermal, and ultrasonic stability, as shown in Fig. 4b and Suppl. Figs. 23 and 24. Compared with the previously reported condensation control strategies, our LCICD strategy demonstrates overwhelmingly collective advantages in terms of sample preparation and applications (Fig. 4c).

According to the mechanism of LCICD, we speculate that if the condensation nucleation activities of substances other than water are sensitive to the liquid-induced change of surface molecular conformation or, more specifically, are distinct on surfaces unshielded and shielded by flexible segments, the condensation rate/density of these substances can be discrepant in area "in" and "out" due to the difference of the favorable nucleation site density. That is, the LCICD effect is not restricted to the water condensation process. As demonstrated in Suppl. Fig. 25, the LCICD effect is also observed during the condensation of acetone, EtOH, n-propanol, and dichloromethane. In brief, the discovery that liquid contact-induced surface adaption affects condensation behaviors offers a convenient strategy for spatial control of condensation, which has been reported to be useful for many applications such as information storage and encryption[6,36], microfluidics[37,38], surface probing[6,39], diffractive optics[7], structural colouration[22], heat transfer[40], anti-icing[19,21], and microreactors[41,42].

## Methods
### Substrate preparation
(1) Crosslinked PDMS. First, the PDMS oligomer (Sylgard 184, Dow Corning) and the curing reagent were thoroughly mixed by a weight ratio of 10:1. Then, the mixture was centrifuged for 10 min at 3257×*g* to deaerate and carefully poured into a petri dish. Finally, it was degassed again under vacuum for 30 min and cured at 80 °C for 2 h. The obtained crosslinked PDMS was ultrasonically cleaned twice with EtOH (5 min for each time). (2) Coverslips anchored with SiO₂ nanoparticles. First, the coverslips were modified with APTES[43]. Briefly, the clean coverslips were hydroxylated by piranha solution[44] and then immersed in the freshly prepared APTES (>98%, TCI Development Co., Ltd.) solution (5 mmol L⁻¹, solvent: mixture of water and acetone with the volume ratio of 1:5) for 30 min. After that, the coverslips were taken out and ultrasonically cleaned with acetone and EtOH for 5 min, respectively. Then, the coverslips covered with APTES were soaked in the SiO₂ aqueous dispersion (1 mg mL⁻¹, hydrodynamic diameter: 12.9 ± 1.0 nm, zeta potential: −38.2 ± 1.6 mV) for 1 h. Finally, the coverslips were taken out and rinsed with ultrapure water.

### Sample preparation
(1) FDTS SAM on various substrates[45]. First, the clean substrates such as silicon wafers (Tianjin Semiconductor Technology Research Institute), coverslips (Jiangsu Shitai Experimental Equipment Co., Ltd.), crosslinked PDMS, and coverslips anchored with SiO₂ nanoparticles were hydroxylated by piranha solution or ultraviolet/ozone cleaner (SD-UV4, Novascan, America) for 20 min. The obtained surfaces are named as substrate−OH (e.g., hydroxylated silicon wafers are named as Si−OH). Then, the substrate−OH was placed in a vacuumized

desiccator containing a vial with 6 μL FDTS (96%, Alfa Aesar). After 12 h at room temperature, the desiccator was moved to an oven and heated for 1 h at 60 °C. Finally, the samples were taken out and ultrasonically cleaned with toluene and EtOH for 5 min, respectively (note: for crosslinked PDMS substrate, ultrasonically cleaned twice with EtOH, 5 min for each time; for coverslips anchored with $SiO_2$ nanoparticles, washed thoroughly with EtOH without ultrasonication, to prevent removal of the anchored $SiO_2$ nanoparticles). (2) Si-GOPTS[46]. Si−OH was placed in a vacuum desiccator containing a vial with 6 μL GOPTS (>96%, TCI Development Co., Ltd.). Subsequently, the vacuum desiccator was heated at 150 °C for 1 h and then at 175 °C for 5 h. Finally, the samples were taken out and ultrasonically cleaned with toluene and EtOH for 5 min, respectively. (3) Si-APTES[43]. The method for preparing Si-APTES was the same as that for preparing the coverslips modified with APTES. (4) Alkanethiol SAMs on Au (Au−C$_{12}$ and Au−C$_1$)[47,48]. The gold-plated silicon wafers (gold layer thickness: 100 nm, Suzhou Silicon Electronic Technology Co., Ltd.) cleaned with piranha solution were immersed in the freshly prepared methyl mercaptan (10 wt% in propylene glycol, Shanghai Macklin Biochemical Co., Ltd.)/1-dodecanethiol (>98%, TCI Development Co., Ltd.) EtOH solution (1 mmol L$^{-1}$) for 24 h under the nitrogen atmosphere. Afterward, the samples were taken out and rinsed thoroughly with EtOH. (5) Au-SH[49]. The gold-plated silicon wafers cleaned with piranha solution were placed in a vacuum desiccator, which was then introduced to $H_2S$ gas carefully (Caution! Prevent $H_2S$ leakage). After 24 h at room temperature, the samples were taken out and rinsed thoroughly with EtOH. (6) Si−C$_1$ and Si−C$_2$[42]. Si−OH was immersed in the freshly prepared trimethoxy(methyl)silane (>98%, TCI Development Co., Ltd.) or ethyltrimethoxysilane (>97%, TCI Development Co., Ltd.) solution (5 mmol L$^{-1}$, solvent: anhydrous toluene) at 105 °C for 12 h under nitrogen atmosphere. Then, the samples were taken out and ultrasonically cleaned with toluene, DMF, and EtOH for 3 min, respectively. (7) Non-crosslinked PDMS brush[50]. Substrates (here, we use coverslips) were first treated with ultraviolet/ozone cleaner for 20 min. Then, the PDMS oligomer was spin-coated onto the coverslips to form a thin film of about 200 μm in thickness. Afterward, the coverslips were placed in closed vials and heated at 100 °C for 24 h. Finally, the coverslips were ultrasonically cleaned with tetrahydrofuran and EtOH thoroughly to remove any unreacted PDMS. Note that all prepared samples were blow-dried with $N_2$ in a Class II Type A2 biosafety cabinet and stored in sealed containers within the biosafety cabinet to avoid contamination unless particularly stated. All prepared samples were evaluated using the water contact angle method to preliminarily examine whether the modifications were successful due to the sensitivity of water contact angles to surface compositions (see Suppl. Table 1 for the water contact angle values).

## Characterizations

The elemental analyses of SAM surfaces were determined by X-ray photoelectron spectroscopy (XPS, ESCALab220i-XL, Thermo Fisher Scientific, America). The surface morphology and roughness of SAMs were characterized using atomic force microscopy (AFM, Multimode 8, Bruker, Germany). The contact angles of macroscopic droplets on surfaces were measured through a contact angle goniometer (JC 2000D 3 M, Zhong Chen, China). The contact angles of condensed microdroplets on surfaces were obtained by using conventional optical microscopy that utilizes focal plane shift imaging[51]. The PMIRRAS data of SAM molecules were collected from a Fourier transform infrared spectrometer (INVENIO R, Bruker) equipped with a photoelastic modulator (PEM-100, Hinds Instruments, Hillsboro, OR) and a liquid-nitrogen-cooled mercury cadmium telluride detector. 86° incidence angle was employed. 2000 scans were collected at 2 cm$^{-1}$ resolution for signal averaging. Each measurement was based on an area of about 5 mm × 8 mm. The surface morphology of coverslips anchored with $SiO_2$ nanoparticles were characterized using scanning electron microscopy (SEM, GAIA3, TESCAN, the Czech Republic). The zeta potential and hydrodynamic diameter of $SiO_2$ nanoparticles were measured by a Malvern Zetasizer (Nano ZS90, Malvern Instruments Ltd., UK).

## Condensation experiment

The condensation experiment was carried out in a closed cell consisting of a rubber O-ring (height 2.0 mm, inner diameter 15 mm) sandwiched between two optical microscope coverslips. Inside the closed cell, 2 droplets of water (0.15 μL per droplet) or other substances (2 μL per droplet for n-propanol, 5 μL per droplet for EtOH, acetone, and dichloromethane) were dropped near the edge of the cell to supply condensing sources. The samples that have been contacted with or without liquid (e.g., $H_2O$, DMSO, DMF, EtOH, and PE) were put into the cell. The entire sample pretreatment and cell preparation procedures were performed in a Class II Type A2 biosafety cabinet to avoid contamination. All the water used in the experiments was ultrapure water with a resistivity of 18.2 MΩ cm provided by a Millipore Milli-Q apparatus and filtered through a 0.22 μm membrane. All the reagents used in the experiments were chromatographically pure unless otherwise specified. Subsequently, the closed cell was placed atop a cryostage (Instec HCS621GXY), which was first heated to 5 °C (for water or n-propanol condensation)/2 °C (for EtOH, acetone and dichloromethane condensation) above room temperature at a rate of 5 °C min$^{-1}$ to induce droplets evaporate and then cooled at a rate of 1.5 °C min$^{-1}$ to facilitate vapor condensate. The condensation process was observed using an optical microscope (Leica DM2700M) equipped with a digital camera (Ando 230).

## Reporting summary

Further information on research design is available in the Nature Portfolio Reporting Summary linked to this article.

## Data availability

The data that support the findings of this study are available from the corresponding author upon request.

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

## Acknowledgements

G.B. was supported by the National Natural Science Foundation of China (Grant No. 22072036), Hebei Natural Science Foundation (Grant No. B2020202034), Central Guidance for Local Scientific and Technological Development Funding Projects (Grant No. 236Z1502G), open research fund of Songshan Lake Materials Laboratory (Grant Number 2022SLABFN03), and "Three Three Three Talent Project" Funding Project of Hebei Province (Grant No. C20221017). D.G. was supported by the National Natural Science Foundation of China (Grant No. 21905072). C.G. was supported by the National Natural Science Foundation of China (22374109 and 21974102) and the National Key R&D Program of China (2018YFA0703700).

## Author contributions

G.B. conceived the project and designed the experiments. H.Z., G.B., H.F., M.R. and Y.L. performed the experiments. G.B., H.Z., D.G., C.G. and M.R. analyzed the data. G.B. and D.G. prepared the paper.

## Competing interests

The authors declare no competing interests.
