## [Peer Review File · Nature Communications]

Controlled condensation by liquid contact-induced adaptations of molecular conformations in self-assembled monolayersEditorial Note: This manuscript has been previously reviewed at another journal that is not operating a transparent peer review scheme. This document only contains reviewer comments and rebuttal letters for versions considered at *Nature Communications*.

Reviewer #1 (Remarks to the Author):

I appreciate all the changes made by the authors and the revised manuscript stands in a better shape. I am happy to recommend the publication of this work after the authors address the following minor comments.

1: The abbreviation of the phenomenon is LCID which refers to liquid contact-induced decrease. I feel the key point of this phenomenon is not captured by this phrase. Better presentation is preferred.

2: One of the key applications is the information storage. Its advantage is described in Fig. 4c, which, actually is elaborated sufficiently. It is still hard to discern the advantage of this method over others.

Reviewer #2 (Remarks to the Author):

As I stated in my original review, I find the macroscopic observations enticing and interesting, but the explanations difficult to follow. The authors have now at least answered my question about the reproducibility, and it turns out that the ratio between the symmetric and asymmetric peaks of the CH₃ band, which is what their conclusions of the monolayer orientation are based on is significantly different between the two parts of the surface that show a different propensity for water nucleation. However the difference is still very small and significance is hard to judge with only three measurements. I recommend publication of this paper in Nature Comms. provided the authors put the original spectra in the supplementary materials so that everybody can judge for themselves whether they find the observed differences significant or not.

Reply to the comments of reviewer #1

I appreciate all the changes made by the authors and the revised manuscript stands in a better shape. I am happy to recommend the publication of this work after the authors address the following minor comments.

Reply: We thank the reviewer's careful reviewing and recommending the publication of our work.

1: The abbreviation of the phenomenon is LCID which refers to liquid contact-induced decrease. I feel the key point of this phenomenon is not captured by this phrase. Better presentation is preferred.

Reply: LCID was the abbreviation of "liquid contact-induced condensation rate/density decrease" To be more accurate, the abbreviation has been changed as LCICD.

2: One of the key applications is the information storage. Its advantage is described in Fig. 4c, which, actually is elaborated sufficiently. It is still hard to discern the advantage of this method over others.

Reply: It should be noted that the traditional surface condensation control strategies always need surface chemical and topographical heterogenization engineering, which are complicated and difficult to scale up. For example, chemical/topographical heterogenization engineering have to use relatively tedious, time-consuming, and expensive techniques such as ion etching, laser ablation, stamping, selective plasma etch, or photolithographic lift-off to achieve biphilic/topographical pattern. Therefore, one of the most obvious advantages of our method is **the simplicity of sample preparation**, which does not need any tedious, time-consuming, or expensive techniques to construct biphilic/topographical pattern. Just a SAM modification can meet the requirement.

Furthermore, unlike the traditional strategies, for which the samples must be firstly customized according to the targeted condensation pattern and the distribution of

condensation droplets cannot be changed any more once the sample have been customized, for our strategy, the prepared sample is universal and the condensation droplet distribution can be freely customized/refreshed just through the contact pattern of liquid on sample surface. Therefore, the other obvious advantages of our method are (1) **the universality of the prepared samples**: our method achieves the separation of information storage process from the sample preparation process and we need not customize the special samples according to the information to be stored, that is, the simply prepared samples are applicable for any information to be stored; (2) **reusability of the prepared samples**: for the traditional condensation strategies, the customized samples can only display the fixed information pattern that has been stored during the sample preparation; while, for our method, the stored information pattern can be erased and rewritten repeatedly.

In addition, our method can realize the **condensation control of nonaqueous substance, trajectory tracking, reversible information appearance/hiding**, and also possess the **better/comparable tolerance and resolution** compared to the most condensation control strategies.

Therefore, our method demonstrates overwhelmingly collective advantages over other strategies. More discussions have been added in the manuscript.

Reply to the comments of Referee #2:

As I stated in my original review, I find the macroscopic observations enticing and interesting, but the explanations difficult to follow. The authors have now at least answered my question about the reproducibility, and it turns out that the ratio between the symmetric and asymmetric peaks of the CH₃ band, which is what their conclusions of the monolayer orientation are based on is significantly different between the two parts of the surface that show a different propensity for water nucleation. However the difference is still very small and significance is hard to judge with only three measurements. I recommend publication of this paper in Nature Comms. provided the authors put the original spectra in the supplementary materials so that everybody can judge for themselves whether they find the observed differences significant or not.

Reply: Thanks for the publication recommendation. The original spectra have been added in Supplementary Fig. 17.